# Temperature Stress Analysis of Super-Long Frame Structures Accounting for Differences in the Linear Expansion Coefficients of Steel and Concrete

Yigang Jia [1,2], Liangjian Lu [1], Guangyu Wu [1,2,*], Bo Zhang [1] and Huibin Wang [1,2]

[1] School of Civil Engineering and Architecture, Nanchang University, Nanchang 330031, China; jiayigang999@sina.com (Y.J.); 351113619005@email.ncu.edu.cn (L.L.); zhangbo_tujian@nerin.com (B.Z.); wanghuibin2021@sina.com (H.W.)

[2] Design and Research Institute, Nanchang University, Nanchang 330031, China

* Correspondence: wuguangyu@ncu.edu.cn; Tel.: +86-189-0708-1115

**Abstract:** Temperature stress analysis is of prime importance to ensure the adequate servicing of super-long frame structures during their service life. The existing design codes and recommendations for reinforced concrete (RC) structures provide methodologies for the reinforcement of design elements that neglect differences in the linear expansion coefficients of steel and concrete. In this paper, we present a numerical method based on a degenerated three-dimensional solid virtual laminated element for simulating and analyzing the temperature stress of a two-layer super-long frame reinforced concrete structure subjected to allover cooling action, whereby the difference in the linear expansion coefficients of steel and concrete are taken into consideration. The results show that the difference in the linear expansion coefficients of steel and concrete, with different constraints, affects the temperature stress experienced by each material in the structure, and this difference adversely affects attempts to avoid structure cracking.

**Keywords:** super-long frame structure; temperature stress; linear expansion coefficient

## 1. Introduction

The accurate prediction of stress levels caused by temperature effects is the main factor in the correct design of structures [1]. This is especially the case for super-long frame structures, which are designed without shrinkage and tension joints on the basis of building aesthetics and structure integrity. As a result, temperature stress can manifest in complex and unexplained ways [2–10]. Therefore, reasonable measures to analyze and control temperature stress are necessary, and research on the variation and distribution of the temperature stress of super-long frame structures under the action of temperature variations is of great significance.

In the past, the temperature stress analysis of super-long structures has considered mixed steel and concrete structures as a single homogeneous structure and neglected the difference in the linear expansion coefficients of steel and concrete [11]. However, the relevant literature on this subject indicates that this difference should be considered in the temperature stress analysis of members and structures. Wang [12] concluded that this difference in the linear expansion coefficients of steel and concrete should be considered since it can cause appreciable temperature stress of a structure with a high reinforcement ratio or under high temperatures, whereas Long [13,14] tested the temperature stress caused by this and concluded that it should be considered in the analysis of concrete structures with crack restriction requirements. Zhang [11] established a calculation model and differential balance equation for axial tension members and concluded that the difference in the linear expansion coefficients of steel and concrete can increase the tensile stress of concrete. Wu [15] carried out a temperature test of small concrete specimens and analyzed the influence of this difference in linear expansion coefficients of steel and concrete on concrete

crack development. Gong [16] analyzed the influence of this difference on reinforced concrete hydraulic structures and concluded that it leads to the inconsistency between the design stress and the actual stress and should be taken seriously by the engineering community.

In spite of these findings and recommendations, the difference in linear expansion coefficients of steel and concrete has rarely been studied in the temperature stress analysis of super-long structures [2–10]. Due to the intrinsic difficulty of temperature analysis and the inability of simulation methods, designers and researchers have analyzed the temperature stress of super-long structures in a simplified manner and neglected the difference in linear expansion coefficients of steel and concrete. In the Chinese design code for concrete structures [17], designers are required to reasonably set the spacing of shrinkage and tension joints through the effective analysis of various adverse factors.

This paper proposes a numerical method based on the degenerated three-dimensional solid virtual laminated element [18] that can simulate the difference in linear expansion coefficients of steel and concrete. In order to understand the influence of the difference in linear expansion coefficients of steel and concrete on the temperature stress of super-long frame structures, a two-layer super-long frame structure simulation model is adopted in this paper and applied in the analysis of temperature stress in a super-long frame structure subjected to allover cooling action. Using this approach, the temperature stress of the members in different positions and their different sections are compared and analyzed, and the information is used to evaluate the influence of the difference in linear expansion coefficients of steel and concrete on the temperature stress in a super-long frame structure.

## 2. Analysis Methods and Calculation Cases

### 2.1. Brief Introduction of the Adopted Degenerated Three-Dimensional Solid Virtual Laminated Element and Analysis Methods

The degenerated three-dimensional solid virtual laminated element [18] introduces the concept of a virtual node and defines a non-real node capable of describing geometrical form and deformation features and establishes a finite element model which can describe structural geometry and stress characteristics by using the laminated beam and degenerated solid elements of the shell. Based on the theory of the three-dimensional solid isoparametric element, the degenerated three-dimensional solid virtual laminated element is acquired by adopting the elastic coefficient matrix modification method and limiting relative displacement, directly introducing a range of basic hypotheses, such as for the beam, plate, shell, membrane, etc. This element has only a linear degree of liberty regardless of the angular degree of liberty and is capable of making a thin connection between the elements and considering the shear effects in every direction. In addition, these elements can be used to precisely analyze the effects of torsion and structure deformation and to avoid a neutral axis and neutral surfaces.

The greatest advantage of the element is that it allows different types of 3D entities in the form of degenerated elements and materials to exist within a finite element. The finite element is segmented into different blocks. The geometric form of each block consists of a hexahedron with ca. 8–20 nodes. The geometrical information and material attribute of each block can be separately defined. The element can be divided according to the characteristics of beam, plate, column, and wall, in which steel and concrete with different linear expansion coefficients and temperatures can be assigned.

The analysis method used in this study is based on the consideration of geometric nonlinearity and material nonlinearity. The T.L. method [19] is used to calculate the geometric nonlinearity of the structure; the reinforcing steel bar features the dispersed reinforcing steel bar (the three-node one-dimensional isoparametric element is used to describe the reinforcing steel bar, and the reinforcing steel bar is regarded as a block in the entity degenerated element. The concrete elastic–plastic constitutive model adopts the multiple intensified plastic model of Ohtani and Chen [20]; while the orthogonal distribution crack model is adopted for concrete cracking.).

The degenerated three-dimensional solid virtual laminated element method is well-suited for analyzing the temperature effects of frame structure taking into account differences in the linear expansion coefficients of steel and concrete.

*2.2. Calculation Case Analysis and Test Verification*

Reinforced concrete (RC) specimens tested under high temperature by Long et al. [14] were selected and analyzed to illustrate the capability and accuracy of the present FE model. Since its results have been reported in detail, this test was selected to facilitate FE simulations and detailed comparisons.

The RC specimens had a span of 900 mm, a width of 150 mm, and a depth of 150 mm and were constructed with a normal strength concrete class (C20). Reinforcement steel was considered to be of class HRB335. The RC specimens were divided into two groups according to the reinforcement ratios of 1.4% and 2.18%. The RC specimens were maintained under natural curing conditions for 28 days and were subsequently stood for 10 months to reduce the effects of contraction. The RC specimens were heated in the test apparatus under the temperature conditions of 20, 40, and 60 °C, respectively. The specific test methods and procedures are described in [14], in which the displacement and the reinforcement stresses were measured considering the difference in the coefficient of linear expansion between steel and concrete.

Due to the discreteness of the test results, some stable results were selected for comparative analysis based on the overall trend of the test results. Table 1 present comparisons between the test results and the FE predictions for the RC specimens. The close agreement between the predictions and the test results demonstrates the validity and accuracy of the proposed FE model. Thus, it is thought that the degraded three-dimensional solid virtual laminated element method and the finite element model based on the method can be used for the simulated temperature stress analysis of a super-long frame structure which considers differences in the linear expansion coefficients of steel and concrete.

**Table 1.** The comparisons between the test data and the FE predictions.

| | Reinforcement Ratio (%) | Elongation per Unit ($10^{-4}$) | | | Reinforcement Stress (Mpa) | | |
|---|---|---|---|---|---|---|---|
| | | 20 °C | 40 °C | 60 °C | 20 °C | 40 °C | 60 °C |
| Test data | 2.18 | 2.0352 | 4.1300 | 6.2550 | −6.8323 | −13.6646 | −20.4968 |
| | 1.4 | 2.0260 | 4.0630 | 6.2220 | −7.2088 | −14.4175 | −21.6263 |
| FEM | 2.18 | 2.0827 | 4.1657 | 6.2519 | −6.4341 | −12.8680 | −19.2710 |
| | 1.4 | 2.0582 | 4.1165 | 6.1752 | −6.9189 | −13.8380 | −20.7570 |
| Ratio(%) | 2.18 | 2.34 | 0.86 | −0.05 | −5.83 | −5.83 | −5.98 |
| | 1.4 | 1.59 | 1.32 | −0.75 | −4.02 | −4.02 | −4.02 |

## 3. Super-Long Frame Structure Analysis Model

*3.1. Structural Model Design*

According to [21], the restraint of the column on the beam and the slab weakens rapidly with the increase in the distance between the joints and the ground. Hence, the actual multi-story super-long structure can be simplified into a two-layer super-long structure in the analysis of temperature effects. To evaluate the performance of the proposed digital framework, a two-layer super-long frame reinforced concrete structure, where each layer has a height of 3 m, is designed to comply with the Chinese design code for concrete structures [17]. Figure 1 shows the structure configuration, geometric cross-section size and reinforcement details. Material properties are provided in Table 2.

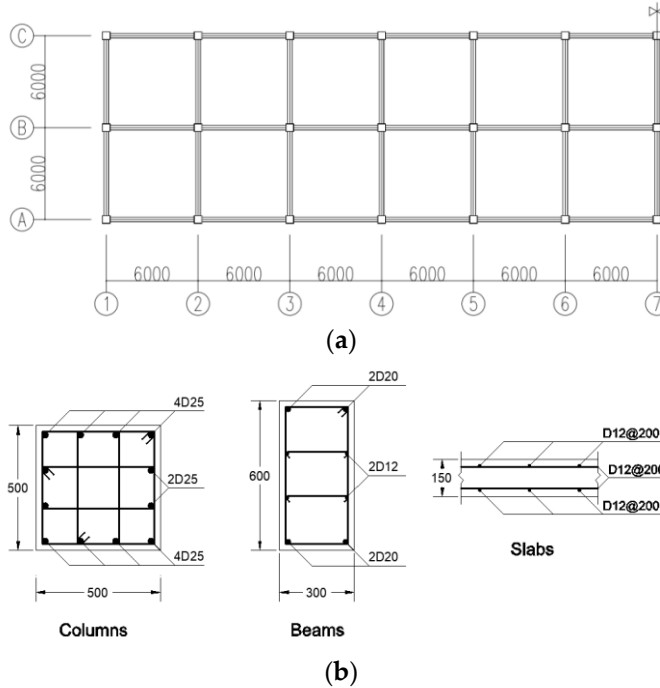

**Figure 1.** Design of the reinforced concrete structure. (**a**) The structure configuration (unit: mm). (**b**) The geometric cross-section size and reinforcement details (unit: mm).

**Table 2.** Material properties.

| Concrete (C35) | |
| --- | --- |
| Density ($\rho$c) | 2600 kg·m$^{-3}$ |
| Poisson's ratio ($\nu$c) | 0.2 |
| Yong's modulus (Ec) | 31.5 GPa |
| Aver. comp. strength (fck) | 23.4 MPa |
| Aver. tensile strength (fctk) | 2.2 MPa |
| Linear expansion coefficient ($\alpha$c) | 0.00001 °C$^{-1}$ |
| **Steel (HRB400)** | |
| Density ($\rho$s) | 7800 kg·m$^{-3}$ |
| Poisson's ratio ($\nu$s) | 0.3 |
| Yong's modulus (Es) | 200 GPa |
| Aver. yield strength (fsyk) | 400 MPa |
| Linear expansion coefficient ($\alpha$s) | 0.000012 °C$^{-1}$ |

### 3.2. Modeling and Simulated Analysis

The finite element model based on the degenerated three-dimensional solid virtual laminated element method was established according to the design of the two-layer super-long frame structure, and this model can simulate the temperature action and the difference in linear expansion coefficients of steel and concrete, as shown in Figure 2.

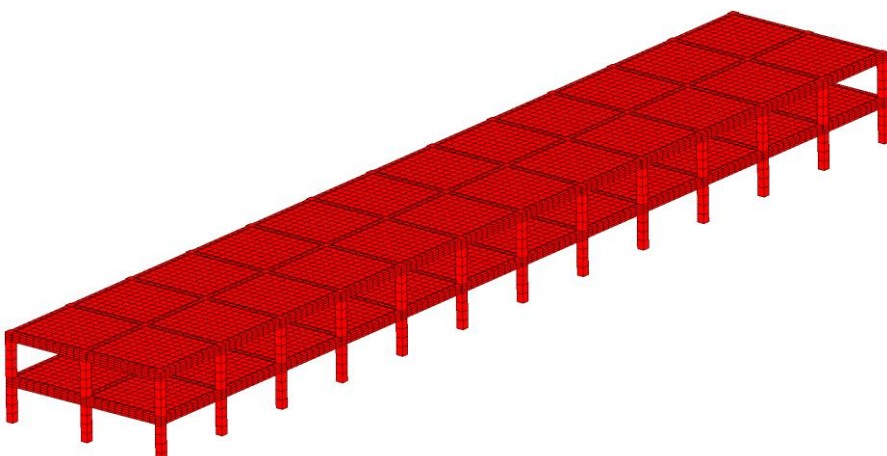

**Figure 2.** Vertical view of the finite element mesh.

The tensile strength of concrete is generally much smaller than its compressive strength, as long as the temperature stresses caused by the cooling effect are verified in the structure. In order to analyze the temperature stress in the super-long frame structure specifically caused by the difference in linear expansion coefficients of steel and concrete, the structure is only subjected to self-weight load and cooling effect. Two working conditions are used in the present document to assess the performance of the structure, namely working conditions one and two. Working condition one considers only uniform cooling at 30 °C without including the difference in linear expansion coefficients of steel and concrete, whereas working condition two includes both uniform cooling at 30 °C and the difference in linear expansion coefficients of steel and concrete.

## 4. Calculation Results and Analysis

The results of model analysis are addressed in this section, with focus on the findings obtained under the two different working conditions and on the main differences with the location of the members in the super-long frame structure. In order to facilitate the understanding of the results presented here, some key points from the analysis are presented in Figure 3.

Figure 3a shows the overall displacement of the super-long frame structure under uniform cooling at 30 °C, whereas Figure 3b shows the specific location of the members which are in the position of the largest structural displacement, and these members are selected to carry out temperature stress analysis. Figure 3c shows the specific location of some key points of the analysis: (i) the key points of the four columns (numbered from column one to column four) are distributed along the four column ridges which are numbered from line one to line four; (ii) the key points of the six beams (numbered from beam one to beam six) are distributed along the middle lines of the bottom and top of the six beams. Among them, the beams in the brackets are indicated in the second layer of the structure; (iii) and the key points of the three slabs (numbered from slab one to slab three) are distributed along the three paths (numbered from path one to path three) of the bottom and top of the three slabs. Among them, the slab in the bracket is indicated in the second layer of the structure.

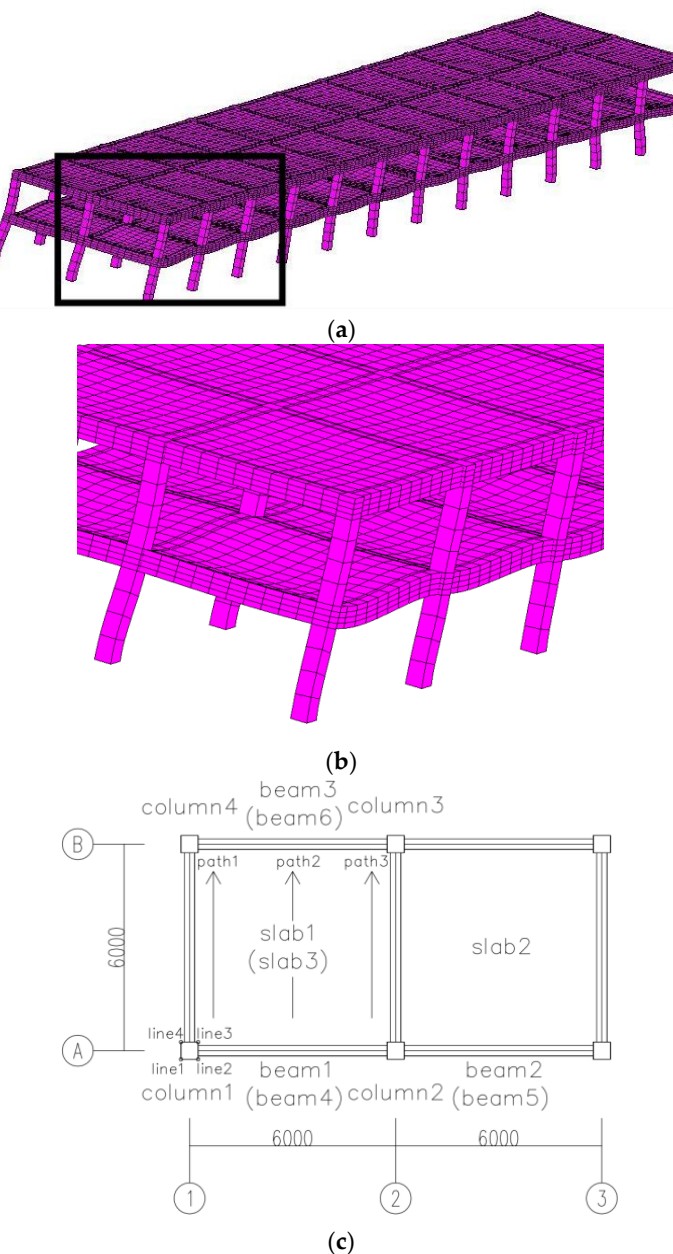

**Figure 3.** View of the structure and the members considered in the discussion of the results. (**a**) The overall displacement of the structure. (**b**) The magnified view of the members with the largest displacement. (**c**) The specific location of the members considered in the discussion of the results.

### 4.1. Temperature Stress Analysis of Columns

The cracking of concrete is the main adverse consequence of overall cooling of the super-long frame structure. Meanwhile, the stress along the column vertical direction is the main factor that causes column cracking. Therefore, this section mainly analyzes and compares concrete stress along the column vertical direction among the columns located in different positions of the structure under the two different working conditions. Figure 4 shows the distribution of the concrete stress along the column vertical direction (Z-axis direction), which include the four lines (numbered from line one to line four) of the four columns (numbered from column one to column four) under the two different working conditions (numbered condition one and condition two). Among them, the data on the X axis represent the temperature stress along the column high direction, and the data on the Y axis represent the distance from the column bottom along the column ridge.

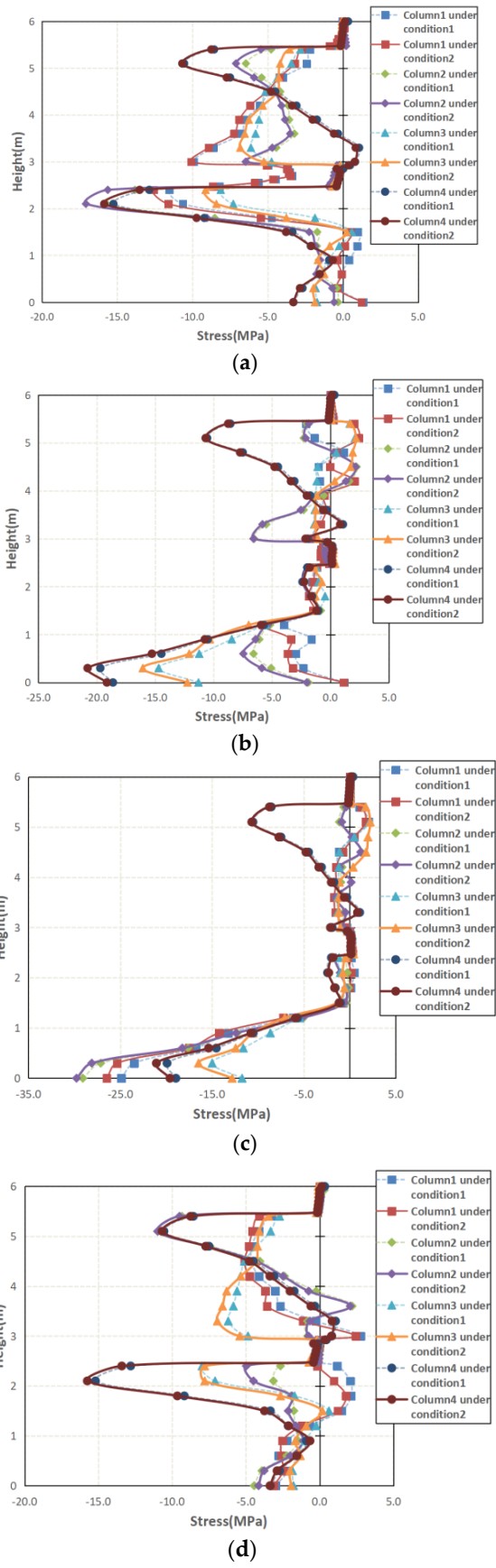

**Figure 4.** Stress profiles over the four columns (numbered from column one to column four) for different column ridges: (**a**) along the ridge numbered line one; (**b**) along the ridge numbered line two; (**c**) along the ridge numbered line three; and (**d**) along the ridge numbered line four.

From Figure 4, it is observed that the difference in the linear expansion coefficients of steel and concrete can influence the temperature stresses of the columns, and the influence varies with the position of the column ridges and the columns in the structure. By comparing and analyzing the four sections of Figure 4, the following important information can be obtained: (i) the stress distribution along the ridges numbered lines one and four obeys a similar law and shows that tensile stress mainly occurs in the first layer of the structure; (ii) the stress distribution along the ridges numbered lines two and three obeys a similar law and shows that tensile stress mainly occurs in the second layer of the structure; (iii) the stress distribution of columns three and four is more regular than that of columns one and two and shows that the amplitude of stress change in column four is greater than that in column three; and (iv) under the working condition two, both lines two and three of column three exhibit significant tensile stress.

From the above information, we can infer that the tensile stress of the column section is greatly affected by the constraints, and that tensile stress is more likely to occur on the side away from the center of the structure in the fixed end or on the side close to the center of the structure in the free end. In addition, the closer the column is to the symmetry axis of the structure, the more regular the stress changes; meanwhile, the further the column is away from the center of the structure, the larger the amplitude of stress change.

Next, the influence of the difference in coefficient of linear expansion between steel and concrete is analyzed by comparing the stresses under the two working conditions (numbered condition one and condition two). Figure 4a shows the difference in the stress on line one of the four columns (numbered from column one to column four) under the two different working conditions in which the largest tensile stress difference occurs in the middle of the first layer of column one. Consideration of the difference in linear expansion coefficients of steel and concrete is more beneficial for avoiding concrete cracking, such that calculating according to working condition one is safe. The same conclusion can be obtained from line four of column one in Figure 4d. However, according to Figure 4b,c, a larger tensile stress appears in the middle of the second layer of column three under working condition two than in working condition one, which adversely affects attempts to avoid concrete cracking.

Therefore, due to the weakening of constraints, the temperature stress analysis of the second layer of columns needs to consider the difference in coefficient of linear expansion between steel and concrete, which can cause internal restraint and lead to greater concrete tensile stress.

### 4.2. Temperature Stress Analysis of Beams

Temperature action is the dominant factor that causes cracking of the super-long frame structure, so the length direction of the structure should be specifically considered [22]. Therefore, this section mainly analyzes and compares the concrete stress along the beam length direction among the beams (as shown in Figure 3c) located in different positions of the structure under the two different working conditions. Figure 5 shows the distribution of concrete stress along the beam length direction (X-axis direction), which includes the bottom and top of the six beams (numbered from beam one to beam six) under the two different working conditions (numbered condition one and condition two). Among them, the data on the X axis of the figure represent the distance from the beam left along the beam length direction, and the data on the Y axis of the figure represent the temperature stress along the beam length direction.

It is observed from Figure 5 that the difference in linear expansion coefficients of steel and concrete can influence the temperature stresses of the beams, and the influence varies with the bottom and top of the beams and the position of the beams in the structure. By comparing and analyzing the two sections of Figure 5, the following important information can be obtained: (i) the trends in stress distribution are clearly opposed along the bottom and top of the beams and show that the beams are always bent; (ii) under working condition one, tensile stress mainly occurs in the second layer of the structure; and (iii) under working

condition two, both the left bottom of beam four and the right top of beam six exhibit significant tensile stress. Figure 5a shows the difference of the stress on the bottom of the six beams (numbered from beam one to beam six) under the two different working conditions (numbered condition one and condition two), in which the largest tensile stress difference occur in across one-quarter of beam four, and the tensile stress under working condition two is 2.53 times that under working condition one. The same phenomenon also appears at the top of beam five and beam six in Figure 5b.

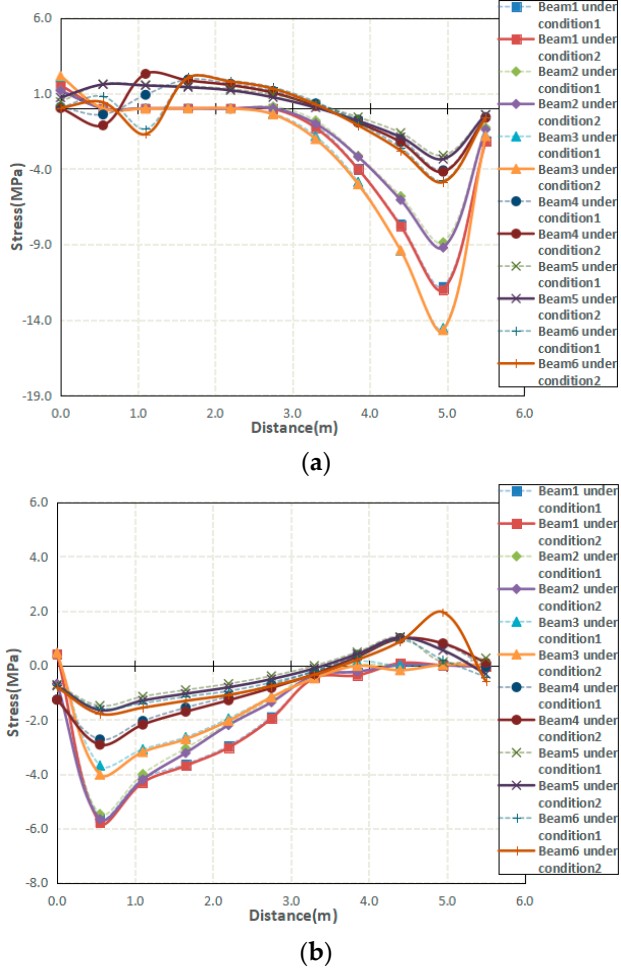

**Figure 5.** Stress profiles over the six beams (numbered from beam one to beam six) for their bottom and top; (**a**) along the middle lines of the bottom; (**b**) along the middle lines of the top.

From the above information, we can infer that the stress distribution of beams is greatly affected by the constraints, and tensile stress is more likely to occur in a weak constraint position. Moreover, the difference in coefficient of linear expansion between steel and concrete can cause greater tensile stress in the weak constraint position than in the strong constraint position. In this case, the difference in the linear expansion coefficients of steel and concrete adversely affects attempts to avoid concrete cracking and should be taken into consideration.

Therefore, due to the weakening of constraints and the strengthening of internal restrain, the second layer of beams are more affected by the difference in the linear expansion coefficients of steel and concrete.

### 4.3. Temperature Stress Analysis of Slabs

The length direction of the structure should be specifically considered for analysis, and the slabs are arranged symmetrically in the width direction of the structure. Therefore,

this section mainly analyzes and compares the concrete stress along the structure length direction among the slabs (as shown in Figure 3c) located in different position of the structure under the two different working conditions. Figure 6 shows the distribution of concrete stress along the structure length direction (X-axis direction), which includes the bottom and top of three paths (numbered from path one to path three) in the three slabs (numbered from slab one to slab three) under the two different working conditions (numbered condition one and condition two). In the figure, the data on the X axis represent the temperature stress along the structure length direction, and the data on the Y axis represent the distance from the lower edge of the slab along the path direction, which are shown in Figure 3c.

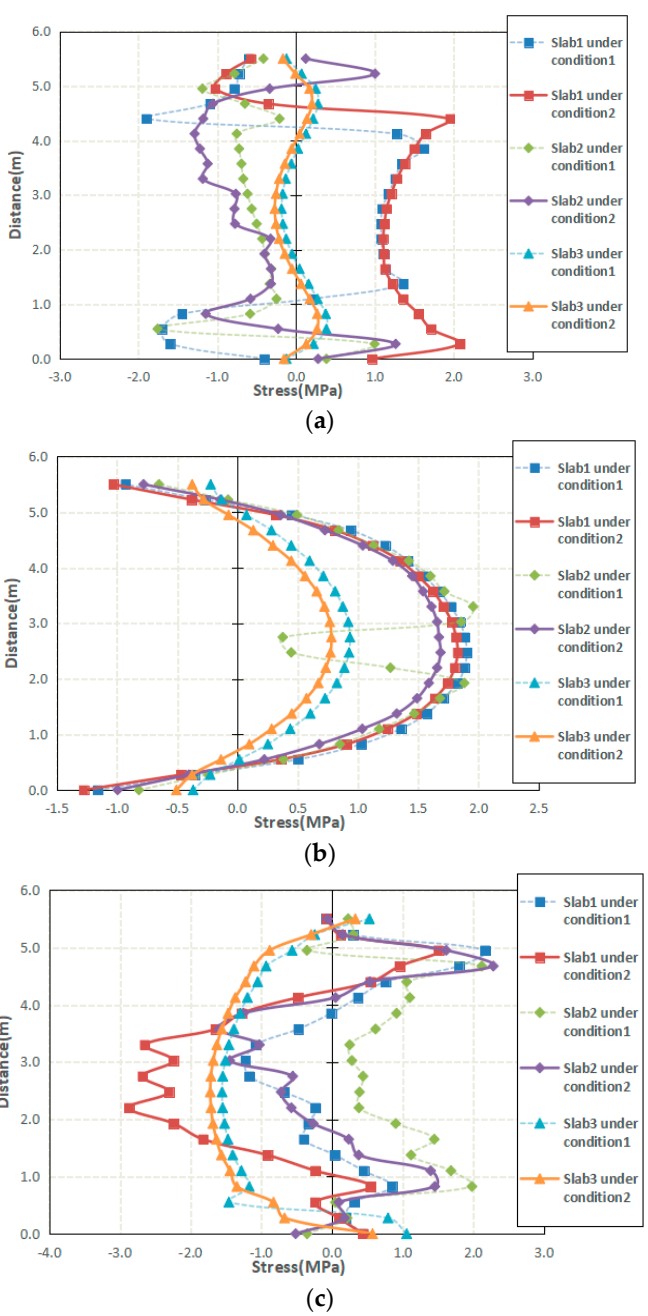

**Figure 6.** *Cont*.

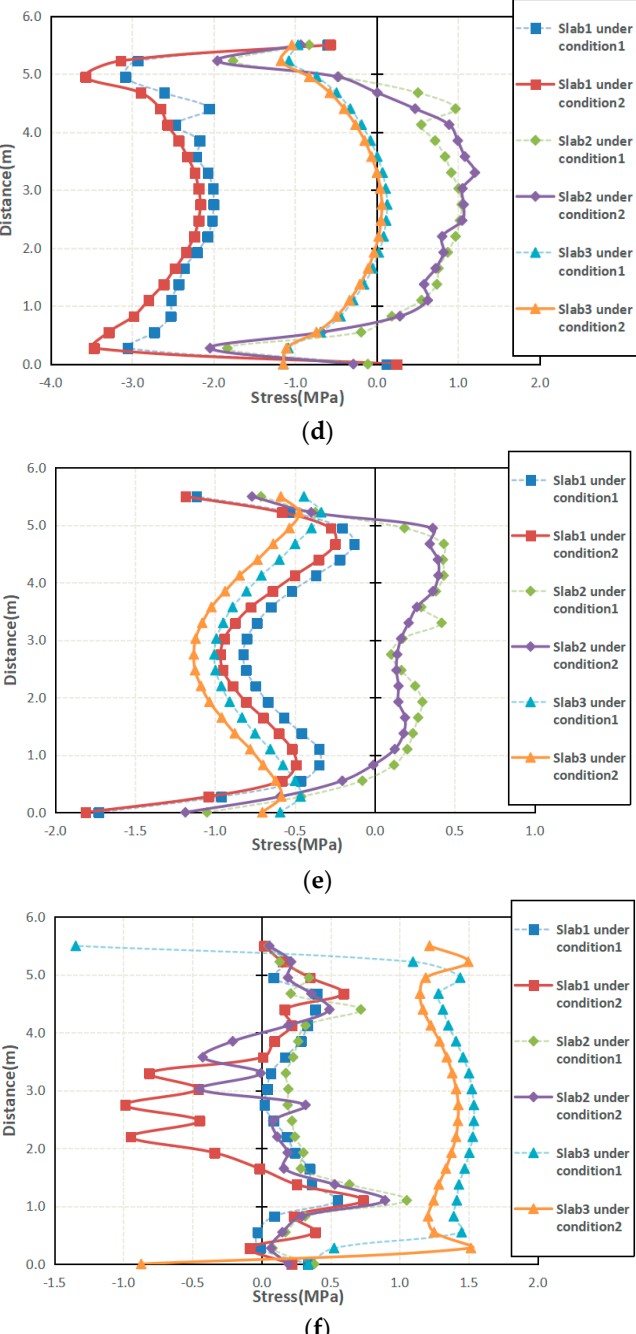

**Figure 6.** Stress profiles over the three slabs (numbered from slab one to slab three) for different paths: (**a**) along the bottom of path one; (**b**) along the bottom of path two; (**c**) along the bottom of path three; (**d**) along the top of path one; (**e**) along the top of path two; (**f**) along the top of path three.

It is observed from Figure 6 that the difference in linear expansion coefficients of steel and concrete can influence the temperature stresses of the slabs, and the influence varies with the bottom and top of the three paths in the slabs and the position of the slabs in the structure. By comparing and analyzing the six sections shown in Figure 6, the following important information can be obtained: (i) the trends in stress distribution along the bottom and top of the slabs are clearly opposed and show that the slabs are always bent; (ii) under working condition one, both the bottom and top of slab two in path two appear to experience different degrees of tensile stress, which indicates that the center of slab two is in eccentric tension; (iii) under working condition two, both the bottom corner of slab one and the top corner of slab three appear to experience significant tensile stress.

From the above information, we can know that the stress distribution of the slabs is greatly affected by the constraints of the surrounding beams, and the end and center of the slabs show different stress states. Moreover, different from the beams, the difference in linear expansion coefficients of steel and concrete can cause greater tensile stress in the position with strong constraints compared to that with weak constraints.

Figure 6a shows the difference in the stress on the bottom of path one in the three slabs (numbered from slab one to slab three) under two different working conditions (numbered condition one and condition two), in which the largest tensile stress difference appears in the corner of the slab one, and the type of stress under working condition two is tensile stress with a value of 2.08 Mpa, and the stress under working condition one is compressive stress with a value of 1.60 Mpa. The difference in the linear expansion coefficients of steel and concrete cause compressive stress to become tensile stress, which adversely affects attempts to avoid slab cracking and must be considered during the temperature stress analysis of the slabs. Figure 6f shows the difference of the stress on the top of path three in the three slabs (numbered from slab one to slab three) under the two different working conditions (numbered condition one and condition two), in which the largest tensile stress difference appears in the corner of the slab three. The tensile stress of the lower right corner under working condition two is 2.89 times that under working condition one, and the tensile stress of the upper right corner under working condition two is 1.36 times under working condition. Therefore, the difference in linear expansion coefficients of steel and concrete adversely affects attempts to avoid concrete cracking and should be taken into consideration. Figure 6b–e also show the change in temperature stress caused by the difference in linear expansion coefficients of steel and concrete, but the change has little effect on the cracking of the slabs.

Therefore, differently from the columns and beams, the first layer and second layer of slabs are both affected by the difference in linear expansion coefficients of steel and concrete. This is due to the fact that the constraints of the slabs are mainly determined by the surrounding beams and are not much affected by the layer of the structure. Moreover, the corners of the slabs are more affected than the center and the edge.

## 5. Conclusions

This paper has presented a super-long frame structure analysis model based on the three-dimensional degenerated virtual laminated element nonlinear finite element method. In the proposed FE model, the difference in linear expansion coefficients of steel and concrete is explicitly considered to enable more accurate analysis of temperature stress. Based on the analysis presented in this work, the following conclusions can be drawn:

(1) In the temperature stress analysis of the super-long frame structure under uniform cooling at 30 °C, the difference in linear expansion coefficients of steel and concrete can lead to variable stress differences in the analysis of members and due to their position in the structure.

(2) Under influence of the difference in the linear expansion coefficients of steel and concrete, a larger tensile stress difference appears in the columns and beams of the second layer compared to those the first layer, due to the weakening of constraints, and this adversely affects attempts to avoid concrete cracking.

(3) Compared to the columns and beams, the slabs are more affected by the difference in linear expansion coefficients of steel and concrete, which can increase the tensile stress and even turn the stress from compressive to tensile in the corners of the slabs, such that the corners of the slabs are more likely to crack than the center and the edge.

(4) The difference in linear expansion coefficients of steel and concrete can cause internal restraint in the super-long frame structure and variation in the temperature stress difference for members in the structure with different constraints, which adversely affects attempts to avoid structure cracking. Therefore, the difference in linear expansion coefficients of steel and concrete should be seriously considered during the analysis and design of super-long frame structures.

**Author Contributions:** Conceptualization, Y.J. and G.W.; methodology, Y.J. and L.L.; software, L.L. and G.W.; validation, B.Z. and H.W.; formal analysis, Y.J., L.L., G.W., B.Z., and H.W.; writing—original draft preparation, Y.J., L.L., and B.Z.; writing—review and editing, L.L., G.W., and H.W.; supervision, G.W. and H.W.; project administration, G.W. and H.W.; funding acquisition, Y.J. and G.W. All authors have read and agreed to the published version of the manuscript.

**Funding:** The authors gratefully acknowledge the research grant provided by the National Nature Science Foundation of China (No. 51268044).

**Institutional Review Board Statement:** Not applicable.

**Informed Consent Statement:** Not applicable.

**Data Availability Statement:** The data used to support the findings of this study are available from the corresponding author upon request.

**Conflicts of Interest:** The authors declare no conflict of interest.

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
