# Peer review of "Temperature Stress Analysis of Super-Long Frame Structures Accounting for Differences in the Linear Expansion Coefficients of Steel and Concrete"

_processes, doi:10.3390/pr9091519_

Round 1
Reviewer 1 Report
As attached

Reviewer 2 Report
Paper is an analysis of the stress state that results from the difference in the thermal expansion coefficient between concrete and steel. The topic is not new. A similar analysis was performed in 1999 by B. Long, S. Li, Y. Sun, and S. Peng (reference [8]). The analysis presented now provides more information.
Practical usefulness of the topic - average. Super long frames are usually designed with thermal dilatations and the stresses are then not so high. However, the construction of long frames without expansion joints is possible and then the thermal stress calculation should also take into account the difference in the thermal expansion coefficient between concrete and steel.
There is no confirmation of the calculation results on a simple experimental model.
In general, I have no other comments. The results of the analysis may be of practical use and I believe that the work should be published.
Author Response
Dear Reviewer:
Thank you very much for your understanding and acceptance.
In spite of some findings and recommendations from members and structures, the difference in linear expansion coefficients of steel and concrete has rarely been studied in the temperature stress analysis of super-long structures. Through the research, we have obtained some useful conclusions which can guide the design and research of super-long frame structures.
The method adopted in this article has been cumulatively implemented and validated in a cumulative manner by this research team in recent years. Because of the cost and the difficulty of the temperature testing of super-long structures, few experimental data can be found. Even so, a test verification about reinforced concrete specimens has been considered in chapter 2.2 to confirm the calculation results on a simple experimental model.
Special thanks to you for your good comments. Those comments are all valuable and very helpful for revising and improving our article, as well as the important guiding significance to our researches. We tried our best to improve the manuscript and made some changes in the manuscript. These changes will not influence the content and framework of the article. We appreciate for your warm work earnestly, and hope that the correction will meet with approval.
Once again, thank you very much for your comments and suggestions.

Round 2
Reviewer 1 Report
Authors modified the manuscript content.